# Performance of a Machine Learning-Based Methicillin Resistance of *Staphylococcus aureus* Identification System Using MALDI-TOF MS and Comparison of the Accuracy according to SCC*mec* Types

**DOI:** 10.3390/microorganisms10101903

**Published:** 2022-09-25

**Authors:** Kibum Jeon, Jung-Min Kim, Kyoohyoung Rho, Seung Hee Jung, Hyung Soon Park, Jae-Seok Kim

**Affiliations:** 1Department of Laboratory Medicine, Hallym University Hangang Sacred Heart Hospital, Seoul 07247, Korea; 2Department of Laboratory Medicine, Kangdong Sacred Heart Hospital, College of Medicine, Hallym University, Seoul 05355, Korea; 3Nosquest Inc., Seongnam-si 13494, Korea

**Keywords:** MRSA, SCC*mec*, MALDI-TOF MS, machine learning model

## Abstract

The prompt presumptive identification of methicillin-resistant *Staphylococcus aureus* (MRSA) using matrix-assisted laser desorption ionization time-of-flight mass spectrometry (MALDI-TOF MS) can aid in early clinical management and infection control during routine bacterial identification procedures. This study applied a machine learning approach to MALDI-TOF peaks for the presumptive identification of MRSA and compared the accuracy according to staphylococcal cassette chromosome *mec* (SCC*mec*) types. We analyzed 194 *S. aureus* clinical isolates to evaluate the machine learning-based identification system (AMRQuest software, v.2.1, ASTA: Suwon, Korea), which was constructed with 359 *S. aureus* clinical isolates for the learning dataset. This system showed a sensitivity of 91.8%, specificity of 83.3%, and accuracy of 87.6% in distinguishing MRSA. For SCC*mec* II and IVA types, common MRSA types in a hospital context, the accuracy was 95.4% and 96.1%, respectively, while for the SCC*mec* IV type, it was 21.4%. The accuracy was 90.9% for methicillin-susceptible *S. aureus*. This presumptive MRSA identification system may be helpful for the management of patients before the performance of routine antimicrobial resistance testing. Further optimization of the machine learning model with more datasets could help achieve rapid identification of MRSA with less effort in routine clinical procedures using MALDI-TOF MS as an identification method.

## 1. Introduction

Antimicrobial resistance, which is the ability of a microorganism to resist the cytotoxic effects of antibiotics, has been observed with increasing frequency over the past several decades [1]. The World Health Organization (WHO) has described antimicrobial resistance as one of the serious threats to global public health [2]. In the hospital setting, the inappropriate and prolonged use of antimicrobial drugs is likely the main contributor to the emergence and spread of highly antibiotic-resistant bacterial infections. In addition, the hospital environment is exposed to the risk of patient-to-patient transmission owing to the presence of highly susceptible immunosuppressed patients and fragile elderly patients as well as other factors such as surgical procedures, clinical therapy intensity, hospitalization length, and failure of infection control [3]. The emergence of antimicrobial resistance presents challenges for both healthcare and socioeconomics [4]. Based on scenarios of the increase in drug resistance by 2050, O’Neil et al. [5] estimated that, unless action is taken, the burden of deaths from antimicrobial resistance could balloon to 10 million lives each year and cost more than $100 trillion in economic losses. Antimicrobial resistance has been observed in most bacteria but is particularly problematic in terms of hospital-acquired infections from multidrug-resistant ESKAPE (*Enterococcus faecium, Staphylococcus aureus, Klebsiella pneumoniae, Acinetobacter baumannii, Pseudomonas aeruginosa,* and *Enterobacter species*) pathogens [6].

*S. aureus* is a Gram-positive, nonmotile, coagulase-positive coccoid bacterial species, which frequently causes hospital and community-acquired infections, skin and soft tissue infections, and pneumonia worldwide [7]. Owing to its intrinsic virulence, capacity to adapt to distinct environmental conditions, and ability to cause a wide array of infections, *S. aureus* is a pathogen of great interest and has long been a threat in hospitals as a common cause of healthcare-associated pneumonia and bloodstream infections [8]. Methicillin-resistant *S. aureus* (MRSA) has been the most prevalent antimicrobial-resistant bacteria since it was first observed in 1961. Through various mechanisms, MRSA has resistance to numerous classes of antibiotics such as β-lactams, cephalosporins, glycopeptides, aminoglycosides, macrolides, and fluoroquinolones. The numerous antibiotic resistance mechanisms that have resulted in the evolution of MRSA limit the choice of antibiotics in clinical practice. Therefore, MRSA infection is difficult to treat, results in significant morbidity and mortality, and requires second-line antibiotics such as vancomycin or new options such as linezolid and daptomycin, which are less effective, more expensive and require careful monitoring to avoid side effects [3]. Consequently, *S. aureus* was selected by the WHO as one of the priority pathogens for which new antibiotics are urgently needed [2]. Therefore, prompt presumptive identification and detection of MRSA is important for the accurate and rapid treatment of patients and infection control in hospitals.

Matrix-assisted laser desorption ionization-time of flight mass spectrometry (MALDI-TOF MS) has become a routine bacterial identification method in recent years [9]. MALDI-TOF MS identifies microorganisms by characterizing the mass spectral composition of a single bacterial colony and enables accurate identification, usually within 24 h after sample collection [10,11]. Conventional culture-based antimicrobial susceptibility methods are time-consuming: the time from sample collection to susceptibility reporting can take up to 72 h. This lengthy reporting of antimicrobial susceptibility risks exposing the patient to inadequate treatment for a significant period of time. Polymerase chain reaction (PCR)-based methods—quantitative PCR (qPCR), reverse transcription quantitative PCR (RT-qPCR), droplet digital PCR (ddPCR), and modified 16S sequencing—take only a few hours and are high-performance and efficacious. However, although such molecular methods have the advantage of being able to report antimicrobial resistance much faster than conventional culture-based methods, they usually include narrow-spectrum assays of single gene targets and are associated with problems relating to specificity of resistance, high labor intensity, and high cost [12]. Recently, a novel antimicrobial susceptibility method based on the detection of nanometric scale oscillations is being developed. It can replace conventional methods due to its rapidity to obtain reliable results in approximately 1–2 h, but is still pre-commercial, in the development stage, and expensive [13]. MALDI-TOF MS is not only capable of simple identification of infectious pathogens but also of performing antimicrobial susceptibility testing using additional information directly from the acquired MALDI-TOF mass spectral data. Since susceptibility testing by MADI-TOF MS was first reported in 2000 [14], MALDI-TOF mass spectra of MRSA have been widely evaluated in several studies, including analysis of single peaks, clusters, and whole spectra [15,16]. Although it is not yet routinely used in clinical practice, direct identification of MRSA using MALDI-TOF MS will be of great benefit in patient treatment and infection control due to its rapid and low-cost advantages.

Methicillin resistance in *S. aureus* is characterized by the *mec*A gene encoding the enzyme penicillin-binding protein 2a (PBP2a), which confers methicillin resistance, causing MRSA to spread worldwide [17,18]. The *mec*A gene is carried in a mobile genetic element called staphylococcal cassette chromosome *mec* (SCC*mec*). To date, 14 SCC*mec* types (I–XIV) with different clonalities have been reported based on the combination of the *ccr* complex and class of *mec* gene complexes [19]. MRSA clones with different SCC*mec* types have evolved distinctively, reflecting diverse lineages [20,21]. These differences could result in different MALDI-TOF peak profiles, and the classification of *S. aureus* strains has been applied in many studies [16,22,23]. Among the *S. aureus* strains, MRSA strains with different SCC*mec* types and methicillin-susceptible *S. aureus* (MSSA) strains can be classified according to their complicated MALDI-TOF MS peak profiles. 

In the last few decades, machine learning has been deployed in various fields as an effective prediction tool that can improve the prediction accuracy of such complicated models. Several studies have used machine learning to identify MRSA using MALDI-TOF MS, but the results varied depending on the model [12,22,24]. In addition, a few studies have compared the performance of machine learning based on SCC*mec* types. If the results of the MRSA identification by MALDI-TOF mass spectra using machine learning algorithm can be evaluated and supplemented according to the SCC*mec* types, MALDI-TOF MS can be used as a more reliable diagnostic tool.

In this study, we applied a machine learning approach to the presumptive identification of MRSA using MALDI-TOF spectra and evaluated the performance of machine learning models according to the SCC*mec* types of MRSA, reflecting the clonal differences.

## 2. Materials and Methods

### 2.1. Clinical S. aureus Isolates

As the test dataset, 194 *S. aureus* clinical isolates isolated at Kangdong Sacred Heart Hospital (Seoul, Korea) from January to December 2018 were included. *S. aureus* isolates were identified using a MicroScan Walkaway 96 Plus (Siemens: West Sacramento, CA, USA). Among the 194 isolates, 36 were isolated from blood cultures and 158 from diverse non-blood specimens including sputum (n = 81, 51.3%), ear swab (n = 16, 10.1%), pus (n = 13, 8.2%), nasal swab (n = 9, 5.7%), soft tissue (n = 9, 5.7%), hemovac tip (n = 8, 5.1%), urine (n = 7, 4.4%), bronchial washing (n = 3, 1.9%), tracheal aspiration (n = 3, 1.9%), throat swab (n = 3, 1.9%), ascitic fluid (n = 2, 1.3%), pleural fluid (n = 1, 0.6%), bronchoalveolar lavage (BAL) fluid (n = 1, 0.6%), bile juice (n = 1, 0.6%), and others (n = 1, 0.6%). In blood culture specimens, positive culture results were detected using an automated detection system (BD BACT/ALERT 3D, Becton Dickinson: Sparks, MD, USA). Isolates were drawn from positive blood culture bottles and spread onto blood plate agar (Becton Dickinson: Sparks, MD, USA) for subculture. Sputum specimens with acceptable quality were used [25]. Urine specimens were inoculated using a 1 μL loop on blood agar plate. For swab specimens collected from wounds, 1.2 mL 0.9% saline was used for rinsing. For pus collected from wounds, the specimens were directly dropped onto the agar. *S. aureus* was cultured for 18 h at 37 °C with 5% CO_2_. The isolates were stored frozen at −70 °C until use. All MRSA and MSSA isolates were confirmed by antimicrobial susceptibility tests performed using MicroScan Walkaway 96 Plus (Siemens: West Sacramento, CA, USA). MRSA was defined as *S. aureus* with a minimum inhibitory concentration (MIC) > 2 μg/mL for oxacillin according to the Clinical and Laboratory Standards Institute (CLSI) guidelines (2016 M100S) [26] and the *mec*A gene is also confirmed by SCC*mec* multiplex PCR typing. 

### 2.2. MALDI-TOF MS Analysis

MALDI-TOF analysis for peptide profiling-based identification was performed in positive linear mode using MicroIDSys (ASTA: Suwon, Korea). Analytical measurements of MALDI-TOF MS were conducted according to the manufacturer’s instructions. A single colony from the blood agar plate was directly smeared onto the target MALDI steel plate without protein extraction. The colony was air dried at room temperature to allow co-crystallization. Dried samples were overlaid with 1.5 μL of 70% formic acid and sequentially with 1.5 μL of CHCA matrix solution (saturated solution of α-cyano-4-hydroxycinnamic acid in 50% acetonitrile with 2.5% trifluoroacetic acid) and the target plate was analyzed. The ASTA MicroID Standard (ASTA) was used for instrument calibration. Spectra were obtained in the range of 1960–20,000 mass to charge (*m*/*z*).

### 2.3. AMRQuest Machine Learning Model for the Identification of MRSA

In total, 359 *S. aureus* isolates randomly selected from approximately 30 strains per year from 2005 to 2014 at Kangdong Sacred Heart Hospital (Seoul, Korea) were used as the training dataset. The isolates from the testing dataset were not used in the training dataset. For MRSA identification and differentiation with MSSA, an enhanced random forest machine learning model was used. The enhanced random forest machine learning model is improved compared to the random forest model in terms of prediction accuracy and can incorporate known relationships between variables and the predictors. When *S. aureus* was identified in the MicroIDSys system, the MALDI-TOF acquisition step for raw spectra began automatically using AMRQuest (v.2.1) (ASTA). The spectra were acquired in linear mode in the ion-positive mode at a laser frequency of 60 Hz and mass range of 2–20 kDa. Each spectrum was obtained from 240 laser shots in six regions of each spot. The acquired raw spectra data were subjected to pre-processing (quality control, smoothing, baseline correction, intensity calibration, and peak detection). Thereafter, an adjustment-merging process (spectra alignment, spectra/peak binning) to construct a feature matrix in which rows represented the samples and columns denoted the aligned peaks was performed. An enhanced random forest machine-learning model was constructed from the matrix after variable selection. After baseline correction and intensity calibration, the data in the temporal domain were converted to *m*/*z* ratios, and the mass spectra within the range of 1960 to 20,000 Da were extracted; the scaling equation was used to ensure signal intensity was in the range of 0 and 1.0 (Figure 1). Therefore, the range of MRSA probability scores was set from 0 to 1.0. MRSA was designated with a probability score greater than 0.5, and MSSA was designated with a probability score less than 0.5. 

### 2.4. PCR for the mecA gene and SCCmec types of S. aureus

*S. aureus* DNA was extracted using the HiYield Genomic DNA Mini Kit (Real Biotech Corporation: Banqiao City, Taiwan), according to the manufacturer’s instructions. The SCC*mec* region harboring the *mec*A gene was assessed using two steps of multiplex PCR (QIAGEN Multiplex PCR Master Mix; QIAGEN: Hilden, Germany) and SCC*mec* element type primers [27]. Reaction mixtures contained 5 μL of genomic DNA, 1 μL of Taq DNA polymerase, and 19 μL of the primer/probe and multiplex real-time PCR master mixtures. Thermal cycles were as follows: 50 °C for two minutes and 95 °C for 15 min; 45 cycles of 95 °C for 15 s, 58 °C for 45 s, and 72 °C for 15 s; 95 °C for five minutes; 35 °C for five minutes; and a melting-curve step (35 °C to 80 °C in 0.5 °C increments for five seconds). The first multiplex PCR step identified the *mec*A and *ccr* types. The second multiplex PCR step classified the gene lineages of *mec*A-*mec*I, *mec*A-IS1272, and *mec*A-IS431. SCC*mec* IV and IVA could be distinguished by the size of the PCR amplicon of *mec*A-IS1272 using modified primers.

### 2.5. Statistical Analyses

Statistical analyses were performed using the SPSS statistics program version 24 (IBM Corporation: New York, NY, USA) and R statistical software (version 3.6.3; R Foundation for Statistical Computing: Vienna, Austria). The Mann–Whitney test was applied to compare nonparametric quantitative variables between the two groups, whereas the Kruskal–Wallis test was used to compare more than two groups. Receiver operating characteristic (ROC) curves were used determine the classifying ability of MRSA and area under the curve (AUC) and 95% confidence interval (CI) were calculated. Statistical significance was set at *p* < 0.05. Predictive performance of the machine learning model for MRSA prediction was presented as sensitivity, specificity, positive likelihood ratio (LR+), negative likelihood ratio (LR−), accuracy, and Cohen’s Kappa Coefficient as follows: Sensitivity = (True Positives (MRSA))/(True Positives (MRSA) + False Negatives); Specificity = (True Negatives (MSSA))/(True Negatives (MSSA) + False Positives); LR+ = Sensitivity/(1-Specificity); LR− = (1-Sensitivity)/Specificity; Accuracy = (True Positives (MRSA) + True negative (MSSA))/(True Positives (MRSA) + True Negatives (MSSA) + False Positives + False Negatives). The sensitivity and specificity indicated the proportion of correct predictions for positive (MRSA) and negative (MSSA) samples, respectively [28].

## 3. Results

### 3.1. Characteristics of Staphylococcus Aureus Isolates

Of 194 *S.aureus* isolates, 106 (54.6%) were MRSA and 88 (45.4%) were MSSA. Of the 36 blood culture samples, 13 (36.1%) were MRSA and 23 (63.9%) were MSSA; of the 158 non-blood clinical samples, 93 (58.9%) were MRSA and 65 (41.1%) were MSSA. Of the 106 MRSA isolates, 65 were SCC*mec* type II (61.3%), 14 were type IV (13.2%), 26 were type IVA (24.5%), and one was type V (0.9%). The ratios of SCC*mec* types (II/IV/IVA/V) were similar between blood culture samples (9/1/3/0) and non-blood clinical samples (56/13/23/1).

### 3.2. Accuracy, Sensitivity, and Specificity of AMRQuest (v.2.1) for MRSA Prediction

MRSA were classified by AMRQuest (v.2.1) model (Figure 1). For the classification of MRSA and MSSA, AMRQuest (v.2.1) showed 87.6% accuracy, 91.8% sensitivity (95% CI: 84.5–96.4) and 83.3% specificity (95% CI: 74.4–90.2). AMRQuest (v.2.1) also showed 0.876 (95% CI: 0.821–0.919) of AUC, 5.510 and 0.098 of positive and negative likelihood ratios, 0.0750 of Cohen’s Kappa Coefficient value, respectively (Table 1). In blood isolates, AMRQuest (v.2.1) showed 91.7%, 85.7% (95% CI: 57.2–98.2), 95.5% (95% CI: 77.2–99.9), 0.906 (95% CI: 0.761–0.977), 20.308 and 0.081 of accuracy, sensitivity, specificity, AUC and positive and negative likelihood ratios, respectively, and in non-blood clinical isolate, showed 86.7%, 92.9% (95% CI: 85.1–97.3), 79.7% (95% CI: 68.8–88.2), 0.863 (95% CI: 0.799–0.912), 4.527 and 0.103, respectively.

### 3.3. MRSA Prediction of AMRQuest According to SCCmec Types

For SCC*mec* type II (n = 65) and IVA (n = 26), classification of MRSA by the machine learning model showed accuracies of 95.4% and 96.1%, respectively. SCC*mec* IV strains (n = 14) showed an accuracy of 21.4% and low probability scores below a median of 0.5, sharing some areas with MSSA. The median probability score of SCC*mec* type IV was significantly lower than that of SCC*mec* type II and IVA (*p* < 0.001) but showed no significant difference compared to that of MSSA. One SCC*mec* type V strain was misidentified as MSSA, and MSSA showed an accuracy of 90.9% (Figure 2, Table 2).

Among the blood isolates, all SCC*mec* type II (n = 9) and type IVA (n = 3) isolates were correctly identified as MRSA, whereas SCC*mec* type IV isolates (n = 1) were misclassified as MSSA. For non-blood clinical isolates, the accuracies of SCC*mec* type II (n = 56) and type IVA (n = 23) were 94.6 and 95.7%, respectively. SCC*mec* type IV (n = 13) showed an accuracy of 23.1%. One SCC*mec* type V strain was misidentified as MSSA. 

## 4. Discussion

Along with routine identification of bacterial colonies, presumptive identification and reporting of resistance with acquired MALDI-TOF MS peak profiles may enable faster and higher-throughput testing in clinical laboratories. This can not only accelerate treatment for multidrug-resistant bacterial infections, but also contribute to antimicrobial resistance suppression by reducing the reliance on broad-spectrum antibiotic therapy and unnecessary broad-spectrum antibiotic use and helping accelerate infection prevention measures, such as the isolation or grouping of patients [12].

Several studies have attempted to identify representative peaks of MRSA in MALDI-TOF spectra through visual examination or software [29,30,31]. However, results have shown inconsistencies because the MALDI-TOF mass spectra of highly related strains can show similarities, making it difficult to distinguish between or interpret different strains. In addition, most peaks detected in the MALDI-TOF mass spectrum corresponded to ribosomal proteins. Protein expression differences among the strains, including ribosomal proteins, may also be present in non-ribosomal proteins [32]. Furthermore, MALDI-TOF MS data consist of hundreds or thousands of *m*/*z* ratios per specimen and an intensity level for each *m*/*z* ratio; therefore, these data require complicated data processing. Machine learning methods can be a solution to these problems. An integrated method of combining data from machine learning models is used as an important and complicated statistical and computational approach in various fields. It can identify important features and predict outcomes by harnessing heterogeneous data [33]. 

In this study, our machine learning model had the ability to classify MRSA by MALDI-TOF MS spectra, with an accuracy of 87.6%, sensitivity of 91.8%, specificity of 83.3%, and positive and negative predictive values of 84.9% and 90.9 %, respectively; the Cohen’s Kappa Coefficient value was 0.750, showing good agreement. Recently, the predictive accuracy of MRSA MALDI-TOF spectra using machine learning was reported to be 77–86.7% [22,24]; our study’s prediction accuracy was 87.6%, showing similar or higher performance. In addition, the sensitivity of our model was higher than that reported by other studies. In a study of five clinics with over 20,000 clinical MSSA and MRSA isolates, Yu et al. [34] reported that the ranges of sensitivity and specificity were 72–83% and 65–88%, respectively. Tang et al. [35] reported a sensitivity of 88.2%, specificity of 90.0%, and accuracy of 88.9% for a total of 224 strains of MRSA and MSSA. 

SCC*mec* types I, II, and III are commonly known to be associated with hospital-acquired MRSA (HA-MRSA) and are multidrug-resistant, whereas SCC*mec* types IV and V are commonly found in community-associated MRSA (CA-MRSA) and tend to be susceptible to most antibiotics other than methicillin and beta-lactam antibiotics [36]. Therefore, knowing the type of SCC*mec* may be important for in-hospital MRSA response strategies. SCC*mec* type IV, the smallest structural and most diverse type, is the most prevalent of CA-MRSA strains worldwide [37,38]. However, in Korea, type IV is rarely reported in relation to SCC*mec* IVA [39,40]. Park et al. [40] reported that in *S. aureus* isolates collected from four regions in Korea in 2007, the rate of IVA type isolates was 23.2%, but that of the IV type was 2.2%. Another study also reported that the SCC*mec* type IVA proportion in CA-MRSA was the predominant type (71.4 %) from 2004 to 2006 in one region of Korea [39]. SCC*mec* type IVA differs from type IV in the presence of a copy of pUB110 [41] and in the remaining areas of SCC*mec*, which are referred to as junkyard (J) regions [42].

In this study, the accuracy of the machine learning model according to type of SCC*mec* showed high performance in type II and IVA (95.4% and 96.1%, respectively), whereas type IV showed a low accuracy of 21.4%. Furthermore, unlike the accuracy for the blood clinical specimen containing only one type IV isolate (accuracy: 91.6%), the accuracy of the non-blood clinical specimen containing all other type IV isolates was as low as 86.7%. Kim et al. [43] reported that SCC*mec* type IV and MSSA have similar genomic characteristics and spectral patterns in MALDI-TOF, and as such, the discrimination between SCC*mec* type IV and MSSA may be challenging. In addition, as mentioned above, the prevalence of SCC*mec* type IV in Korea is generally low, and thus, there may have been limitations from the beginning in the collection of type IV strains for the training dataset in our study as well. Therefore, it is important that MALDI-TOF spectra of more type IV isolates be included in machine model training for better differentiation of type IV and MSSA and better performance. Only one SCC*mec* type V isolate was included in the test dataset, and this isolate was incorrectly identified as MSSA, resulting in an accuracy of 0%. Because the prevalence of SCC*mec* type V in Korea is not high as that of type II and IVA, more isolates are required to be included in the dataset.

In hospital settings, the presumptive reporting of MRSA that could allow for reporting one day earlier than culturing methods could provide useful information for the management of the patient. The positive predictive value of 84.9% was obtained when a specific *S. aureus* isolate reported initially as “presumptive MRSA isolate” was determined to be MRSA with a probability of approximately 85%. Although presumptive reporting is not a standard test for MRSA and there is still a 16–24 h wait associated with obtaining the final antimicrobial resistance test results, this presumptive identification method might provide important and timely information for the treatment of critically ill patients infected with *S. aureus*.

Our study had some limitations. Although the strains for the training dataset were collected over more than a 10-year period to cover different clonalities, the types of SCC*mec* in *S. aureus* were not diverse. Since most of the SCC*mec* types in Korea were type II and type IVA, there were a small number of type IV strains in the training set due to the low prevalence of IV among *S. aureus* bacteremia. Therefore, the characteristics of SCC*mec* IV may not have been properly reflected in the training, which seems to have affected the overall accuracy of the machine learning model of AMRQuest (v.2.1.). However, for the invasive MRSA clones causing bacteremia in Korean hospitals, the accuracy of presumptive reporting seems to be acceptable. More diverse clones can improve the presumptive identification of MRSA. In addition, the clear-cut differentiation between MRSA and MSSA might be impossible with this method because the MALDI-TOF peaks reflect the similarity of the highly expressed proteins and not the resistant determinant genes. Therefore, different reports of machine learning models based on the geometric differences and clonal changes in *S. aureus* strains should occur. Continuous monitoring and proofreading of the presumptive reporting of MRSA should be performed to streamline its processes for clinical use. In addition, coagulase-negative staphylococci (CoNS) are also important pathogens in hospital settings, further study might be needed for the differentiation of methicillin resistance among the major CoNS such as *Staphylococcus epidermidis*.

## 5. Conclusions

The machine learning model of MALDI-TOF MS analysis in this study could be useful for the presumptive identification of MRSA and MSSA isolates. Good accuracy was shown for SCC*mec* types II and IVA, which are the dominant clonal types in Korean hospitals. The proportion of SCC*mec* type IV strains could affect the correct identification of MRSA in clinical settings. Follow-up verification and further optimization of the machine learning model with more datasets could help achieve presumptive identification of MRSA with MALDI-TOF MS analysis with reduced effort in routine clinical procedures. 

## Data Availability

Not applicable.

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
