# Peer review of "Performance of a Machine Learning-Based Methicillin Resistance of Staphylococcus aureus Identification System Using MALDI-TOF MS and Comparison of the Accuracy according to SCCmec Types"

_microorganisms, 2022, doi:10.3390/microorganisms10101903_

Round 1

Reviewer 1 Report

The article "Performance of a machine learning-based methicillin resistance 2 of Staphylococcus aureus identification system using MALDI-3 TOF MS and comparison of the accuracy according to SCCmec 4 types" is of great practical importance, since reducing the time for determining sensitivity/resistance to antibiotics is very important for all hospitals. However, there are some minor comments.

1. In the introduction, in addition to PCR, would be better to refer and compare your work with the research of the team led by prof. S. Kasas, who determines antibiotic resistance by atomic force microscopy.

2. It is necessary to indicate the convergence of results in the study of the same strain (in multiple measure).

3. Need a technical correction in the captions to the figures.

Otherwise, taking into account the significance of the problem, the article can be recommended for publication in the journal Microorganisms.

Reviewer 2 Report

The manuscript by Kibum Jeon et al. describes the performance of a machine learning system to detect MRSA using MALDITOF.

The manuscript is worthy of revision with the following comments: 

Prefer the passive form.

Numbers less than 12 should be spelled out.

Why did the authors not include ATCC strains to have different profiles not well represented in Korea (in addition to the strain used as a control).

Line 129: what did the authors mean by "acceptable quality"?

Line 130: the quantitative loop means?

Line 137: how did the authors determine MIC (diffusion disk, liquid, ...)? Depending on the answer, the need for double-checking could be emphasized.

Did the authors test their algorithm on Coag-neg staphylococci (even if it was not developed for this application, this information is essential).

Paragraph 2.5: Should the formula be referenced appropriately?

SCCmecTypes: Did the authors detect any non-typeable sccmec types?

Figure 1: An ROC curve would be interesting to determine the appropriateness of the chosen threshold. Please consider

Table 1: The CI95 should be determined.

Table 2: Statistical comparison between detection of sccmec types should be shown. 

Discussion: "et al" should be italicized.

Global: prefer likelihood ratio to predictive value, as the study cannot be considered globally representative.

Round 2

Reviewer 2 Report

The manuscript has been improved according to my previous recommandations